# *Centaurea benedicta*—A Potential Source of Nutrients and Bioactive Components

**DOI:** 10.3390/plants13243579

**Published:** 2024-12-22

**Authors:** Olga Teneva, Zhana Petkova, Ana Dobreva, Anatoli Dzhurmanski, Liliya Stoyanova, Maria Angelova-Romova

**Affiliations:** 1Department of Chemical Technology, Faculty of Chemistry, University of Plovdiv “Paisii Hilendarski”, 24 Tzar Asen Street, 4000 Plovdiv, Bulgaria; zhanapetkova@uni-plovdiv.bg (Z.P.); liliyastoyanova@uni-plovdiv.bg (L.S.); maioan@uni-plovdiv.bg (M.A.-R.); 2Institute for Roses and Aromatic Plants, Agricultural Academy, 49 Osvobozhdenie Blvd., 6100 Kazanlak, Bulgaria; anadobreva@abv.bg (A.D.); anatoli630914@gmail.com (A.D.); 3Tobacco and Tobacco Products Institute, Department of Tobacco and Tobacco Smoke Chemistry, Agricultural Academy, 4108 Markovo, Bulgaria

**Keywords:** *Centaurea benedicta* L., *Asteraceae* family, glyceride oil, fatty acids, biologically active components

## Abstract

The Asteraceae family is a large plant family, with over 1600 genera and 25,000 species, most of which are generally herbaceous plants. This family’s members are widely used in the human diet and medicine. One of the most popular representatives is *Centaurea benedicta* L., known as ‘Blessed Thistle’. It is a famous plant in the herbal world with some medical benefits, such as strong antioxidant and antidepressant effects, with antibacterial and antiseptic properties, a stimulant of appetite, with a good effect on the liver and the secretion of bile juices, etc. Therefore, this work aimed to fully characterize the chemical composition of the seeds of *C. benedicta* introduced in Bulgaria, some of the physicochemical characteristics, as well as the biologically active compounds. The main nutrient in the chemical composition was carbohydrates (68.5%), and half of their quantity was occupied by fibers (32.2%). Total proteins accounted for 16.4%, and the glyceride oil content was rather low—about 11.0%. The main fatty acids identified in the seed oil were linoleic (72.1%) and oleic acids (18.1%), and the amount of the polyunsaturated ones predominated (73.0%). The main lipid-soluble bioactive components were sterols (0.9%), phospholipids (1.9%), and tocopherols (492 mg/kg). β-Sitosterol (59.5%) and stigmasterol (19.4%) were the main sterols, and α-tocopherol (472 mg/kg) predominated in the tocopherol fraction. The major phospholipids were phosphatidylethanolamine (45.4%), followed by phosphatidylinositol (37.1%) and phosphatidylcholine (6.1%).

## 1. Introduction

The Asteraceae family, often known as the ‘sunflower family’, is one of the largest flowering plant families. A lot of well-known species are members of this family, such as sunflower, chicory, coreopsis, lettuce, dahlia, and daisy, as well as several medicinal plants, such as wormwood, chamomile, and dandelion [1]. The plant *Centaurea benedicta* (syn. *Cnicus benedictus* L.), or ‘Blessed Thistle’, is also a typical member of the Asteraceae family and belongs to the group of medicinal plants (herbs). The Asteraceae family is distributed all over the world except for Antarctica, but the homeland of *C. benedicta* is considered to be southern Europe [2]. It is native to the Mediterranean region, from northern Portugal to southern France and east to Iran. It is known in other parts of the world, including North America, as an introduced species and often a noxious weed. It is found in grassy areas and near the sea, on dry, sunny landfills and polluted grounds, on stony and arable soil. It grows at an altitude of up to 1000 m [3].

*C. benedicta* L. is an annual plant that typically grows to a height of 60 cm. Its flowers form a dense flower head, adorned with spiny bracts. The plant’s root is fibrous and whitish, from which several reddish stems emerge. These stems bear thick, lance-shaped leaves, densely covered with hairs, reaching up to 30 cm in length and 8 cm in width [3]. The leaf morphology is consistent with that described for some members of the Asteraceae family: small, spiny, and covered with an indumentum and hairs of varying lengths and colors [4]. The leaf edges are lined with small spines, while the surface features prominent whitish veins and a central one. At the plant’s base, the leaves form a rosette. The flowers, which develop at the top of the stems, are yellow and gathered in dense inflorescences with a diameter of 3–4 cm, known as flower heads, surrounded by numerous spiny bracts. The flowering period extends from May to August. The aroma of the flowers is weak and unpleasant. The aerial parts of the plant, particularly the upper portions, are used in medicine. The young leaves can be consumed raw, and the flower heads harvested before blooming serve as a substitute for artichokes. Occasionally, the root of the herb, when boiled, is also used medicinally. *C. benedicta* is typically collected in summer and dried for further use [3,5]. It is used in traditional medicine for its protective properties (anti-inflammatory, antimicrobial activity, etc.) [6] and for diseases of the liver similar to another member of the Asteraceae family, *Centaurea solstitalis* (used to treat stomach problems, abdominal pain, herpes infections, and the common cold) [7]. *C. benedicta* is known for its lactogenic action as a good stimulator of breast milk. The herb is also used to regulate menstruation—for amenorrhea and painful menstruation. It has a good detoxifying effect due to its ability to purify the blood by inducing sweating and enhancing diuresis [6,7]. *C. benedicta* contains essential oils, tannins, resins, mucous substances, lignans, sugars, and minerals—calcium, potassium, iron, magnesiums and manganese [8]. The plant has been analyzed regarding its antimicrobial and antineoplastic effects [9], as well as some chemical compounds (sesquiterpene lactones, triterpenoids, lignans, flavonoids, tannins, essential oils, phenolic compounds, saponins, alkaloids, starch, glycosides, and coumarins) [9,10]. There is a lack of information about the chemical and lipid composition of *C. benedicta* dwelling in Bulgaria, as well as the presence and amounts of biologically active compounds of the seeds and glyceride seed oil of the species.

Based on the above, the aim of the current study was formed, namely, to investigate the chemical composition of the seeds of *C. benedicta*, some physicochemical characteristics of the isolated glyceride oil, as well as its biologically active complex to evaluate its potential as a valuable and cheap source of phytonutrients.

## 2. Results and Discussion

### Chemical Composition of C. benedicta Seeds

The main characteristics that determine the chemical composition of the seeds of *C. benedicta* were investigated—the content of glyceride oil, total protein content, carbohydrates, crude fiber, and mineral substances for 100 g of seeds and in 1000 seeds (the weight of 1000 seeds was 34.692 g). The energy value was calculated. All data about the chemical composition of the seeds (dry weight) are presented in Table 1.

The oil content of the seeds from *C. benedicta* was determined to be 11.0%. This amount is almost half of what has been reported by other authors for the same plant (25.65%), as well as for safflower seeds (29.39%), milk thistle (19.39%), and cardoon (24.49%) [11]. These variations could be attributed to the different climatic conditions under which the plants were grown, as well as inherent differences in the species. Factors such as the soil quality, temperature, precipitation, and altitude can all influence the biosynthesis of oils in plant cells, leading to substantial differences in the oil content. The protein content of the analyzed seeds of *C. benedicta* was found to be approximately 16%. This result is lower than the reported values for the same plant (22.4%) and safflower (22.8%) but similar to niger seeds (19.3%) [11]. Proteins are crucial for the growth and repair of tissues [12]. The carbohydrate content was found to be about 68–69%, and the amount of crude fiber was also comparatively high at 32.2%. Dietary fibers are essential for digestive health, and the high fiber content in *C. benedicta* seeds makes them particularly beneficial for maintaining healthy digestion [12]. According to Vishwakarma and Dubey [13], the fiber content of some medicinal and aromatic plants can vary widely, from 0.90% in *Moringa oleifera* Lam. Moringaceae (Munga) Leaf to 28.59% in *Marsilea minuta* Marsileaceae (Sunsuniya) Leaf. The ash content of the seeds was 4.1%, which is slightly lower than the mineral content reported for the species *Centaurea karduchorum* Boiss (5.93%) by Tunçtürk et al. [14]. The ash content is an indicator of the total mineral content in the seeds, and these minerals are vital for various bodily functions [12]. The moisture content was found to be 5.4%, which is close to other scientific reports of 7% [11]. The moisture content is critical for determining the storage stability of seeds, with a lower moisture content often resulting in a longer shelf life [12]. Finally, the energy value of the dry weight seeds was calculated to be 439 kcal/100 g, highlighting their potential as a significant energy source in the human diet.

Some physicochemical characteristics of the extracted glyceride oil from *C. benedicta* seeds were measured. The results are presented in Table 2.

The peroxide value (PV) is a crucial indicator of the presence of hydroperoxides in oil, which are formed during oxidation processes. A higher PV signifies a greater extent of oxidation, indicating the potential deterioration of the oil. The examined seed oil of Blessed Thistle exhibited a PV of 2.8 meqO_2_/kg, aligning closely with the findings of Ghiasy-Oskoee and Agha Alikhani [11], who reported PVs of 3.71 meqO_2_/kg for Blessed Thistle, 2.60 meqO_2_/kg for safflower, and 2.77 meqO_2_/kg for milk thistle. This relatively low PV suggests that the Blessed Thistle seed oil is of good quality and has not undergone significant oxidation [15]. The iodine value, another critical parameter, measures the degree of unsaturation in the oil, reflecting the quantity of double bonds present in fatty acids. The iodine value of the examined *C. benedicta* seed oil was determined to be 147.3 g I_2_/100 g, which is notably higher than the values reported by Ghiasy-Oskoee and Agha Alikhani [11] for *C. benedicta* (128.7 g I_2_/100 g) and niger seed oil (132.9 g I_2_/100 g). For other seed oils in the Asteraceae family, the iodine values ranged from 121.5 g I_2_/100 g (milk thistle) to 129.81 g I_2_/100 g (cardoon seed oil). This higher iodine value indicates a greater degree of unsaturation, which is often associated with beneficial health properties and the stability of the oil.

The refractive index of the analyzed sample was measured to be 1.4748, which corresponds exactly with the value reported by Ibrahim et al. [16] for safflower oil (1.4748). The refractive index is indicative of the purity and quality of the oil, as it measures the bending of light as it passes through the oil.

The current work was intended to explore *C. benedicta* seeds as a new source of biologically active constituents. The biologically active complex (unsaponifiable compounds, sterols, phospholipids, and tocopherols) was examined, and the results are given in Table 3.

Previous studies lack information on the content of unsaponifiable matter in *C. benedicta*. However, the investigated species was distinguished by a high content of unsaponifiable matter, at 14.5%. This result is significantly higher than those found in sunflower and safflower seed oils, which contain less than 1.5% unsaponifiable substances [16]. Unsaponifiable matter includes bioactive compounds such as sterols, tocopherols, and hydrocarbons, which contribute to the oil’s therapeutic properties and stability [12].

The total sterol content in *C. benedicta* seed oil was investigated and found to be approximately 1.0%. This value is considerably higher than that found in other vegetable oils from the Asteraceae family, such as sunflower oil (0.24–0.46%) [16]. Sterols play a vital role in reducing cholesterol levels and improving heart health, making this high sterol content noteworthy [12].

The total phospholipid content in the oil was measured at 1.91%, which is significantly higher than that of sunflower oil (~1.0%) [17]. Phospholipids are essential for cell membrane integrity and function, indicating the potential health benefits of *C. benedicta* seed oil.

The total tocopherol content was determined to be 492 mg/kg. While this amount is lower than the tocopherol content for Blessed Thistle (670.43 mg/kg) as reported by Ghiasy-Oskoee and Agha Alikhani [11], it is consistent with the range for sunflower oil (440–1520 mg/kg) [16]. Tocopherols, particularly vitamin E, are powerful antioxidants that protect cells from oxidative damage. The result is also significantly lower than the tocopherol content reported by Peker and Bastürk [18] for oils from other *Centaurea* species, such as *Centaurea balsamita* (1186 mg/kg) and *Centaurea albonitens* (1689 mg/kg). Conversely, Teneva et al. [19] reported a considerably lower tocopherol content for *C. thracica* (260 mg/kg).

The individual tocopherol composition in *C. benedicta* seed oil included α-tocopherol and β-tocopherol as the main representatives. In the investigated oil, α-tocopherol was predominant at 472 mg/kg, followed by minimal quantities of β-tocopherol. These results align with previous reports for isolated oils from *C. albonitens* and *C. balsamita*, where α-tocopherol is also the major component [18]. α-Tocopherol is known for its antioxidant activity, making it a critical component for health benefits [12].

Seventeen fatty acids were identified in the total lipids of *C. benedicta*. The results are shown in Table 4.

Yildirim et al. [20] reported that palmitic acid was the predominant fatty acid in some oils from plants of the genus *Centaurea*. In contrast, Erdogan et al. [21] highlighted linoleic, oleic, and palmitic acids as the major fatty acids in the *Centaurea* species they analyzed. This pattern was also observed in the fatty acid composition of our sample.

The fatty acid composition of *C. benedicta* seed oil was found to be consistent with previous reports, showing linoleic acid (C18:2) as the major fatty acid at 69.4%, oleic acid (C18:1) at 18.72%, and stearic acid (C18:0) at 2.70% [11]. Linoleic acid, an essential polyunsaturated fatty acid, is crucial for human health, contributing to skin integrity and metabolic functions [12]. The high content of linoleic acid in the studied oil of Blessed Thistle (69.4%) is significantly higher than the levels reported by Tekeli et al. [22] for six other *Centaurea* species, which ranged from 29.15% in *C. virgata* to 55.27% in *C. kotschyi* var. *kotschyi*.

Our analysis also showed that the amounts of oleic acid (C18:1) and palmitic acid (C16:0) were higher (18.1% and 5.9%, respectively) than previously reported values. Oleic acid, a monounsaturated fatty acid, is known for its benefits in reducing cardiovascular risks and improving cholesterol levels [12]. The variation in the oleic acid content observed by Tekeli et al. [22] ranged from 7.98% in *C. solstitialis* subsp. *solstitialis* to 28.47% in *C. triumfettii*. Palmitic acid, a saturated fatty acid, while generally considered less beneficial, is still an essential component of biological membranes and is involved in numerous metabolic processes [12]. Tekeli et al. [22] reported that the palmitic acid content varied widely from 11.89% in *C. urvillei* subsp. *urvillei* to 28.41% in *C. virgata*.

The stearic acid (C18:0) content in the analyzed *C. benedicta* oil was found to be 2.6%, which is in line with the value reported for *C. kotschyi* var. *kotschyi*. Stearic acid, a saturated fatty acid, is known to have a neutral effect on blood cholesterol levels compared to other saturated fatty acids [12]. Other identified fatty acids were present in trace amounts, ranging from 0.1% to 0.2%.

In the analyzed seed oil from *C. benedicta*, the total saturated fatty acids (SFAs) accounted for 9.0%, significantly lower than the values reported by Erdogan et al. [21] for some *Centaurea* species, which ranged from 24.61% to 50.92%. This lower SFA content is beneficial from a dietary perspective, as a high intake of saturated fats is associated with an increased risk of heart disease [12]. The polyunsaturated fatty acids (PUFAs) in the oil were found to be 72.4%, and monounsaturated fatty acids (MUFAs) were 18.2%. These values are considerably different from those reported by Erdogan et al. [21], where PUFAs ranged from 12.21% to 20.57% and MUFAs from 3.40% to 37.96%. The high PUFA content is particularly notable, as PUFAs are essential for reducing inflammation and promoting heart health [12].

Overall, the fatty acid profile of *C. benedicta* seed oil indicates a composition rich in beneficial unsaturated fatty acids, making it a promising candidate for dietary and medicinal applications.

The major part of the unsaponifiable matter of *C. benedicta* seed oil consists of sterols. That is why the individual sterol composition was investigated as well. The results are given in Table 5.

In the seed oil of *C. benedicta*, all classes of sterols typically found in vegetable oils were detected, with β-sitosterol being the predominant sterol, comprising nearly 60% of the total sterol content. The other identified sterol components included stigmasterol (19.4%), Δ^5^-avenasterol (13.5%), and campesterol (7.5%). This comprehensive profiling of sterols is significant as it provides insight into the nutritional and potential health benefits of *C. benedicta* seed oil. Sterols are known for their cholesterol-lowering properties and play a crucial role in maintaining cellular structure and function [12].

To the best of our knowledge, there has been no previously reported data on the total sterol content and their individual composition in *C. benedicta* seed oil. This study contributes novel information to the existing body of knowledge, filling a critical gap in the literature. The presence of such a high concentration of β-sitosterol is particularly noteworthy, as it is the most abundant plant sterol and is recognized for its ability to inhibit cholesterol absorption in the human intestine, thus aiding in the reduction of blood cholesterol levels [12].

The findings are further supported by the work of Fayed et al. [23], who reported the presence of β-sitosterol and stigmasterol in the aerial parts of *Centaurea pumilio* L., another species within the same genus. However, their study did not encompass the seed oil, highlighting the uniqueness of the current research. The identification of Δ^5^-avenasterol and campesterol in the seed oil of *C. benedicta* adds additional value, as these sterols have been associated with various health benefits, including anti-inflammatory and anti-carcinogenic properties.

The individual phospholipid composition of *C. benedicta* seed oil was determined, and the results are given in Figure 1.

The information available on the total phospholipids and their individual composition in *Centaurea* seed oil is extremely limited. In the analyzed glyceride oil from *C. benedicta*, all classes of phospholipids typical of vegetable oils were identified, including phosphatidylcholine, phosphatidylinositol, phosphatidylethanolamine, phosphatidylserine, phosphatidic acid, and lysophosphatidylcholine. Among these, phosphatidylethanolamine (45.4%) and phosphatidylinositol (37.1%) were the predominant components in the phospholipid fraction, while the others, such as phosphatidylcholine, phosphatidylserine, and phosphatidic acid, were present in similar quantities, ranging from 4% to 6%.

The high content of phosphatidylethanolamine and phosphatidylinositol is particularly notable. Phosphatidylethanolamine plays a crucial role in membrane fusion and the stabilization of membrane curvature, while phosphatidylinositol is involved in signal transduction processes within cells [12].

Significant differences were observed when comparing the individual phospholipid quantities in *C. benedicta* with those reported for *C. thracica* by Teneva et al. [19]. For instance, in *C. thracica*, phosphatidylcholine constituted 9.1%, phosphatidylinositol 25.0%, and phosphatidylethanolamine 31.8%, while phosphatidic acid was not identified. This stark contrast highlights the unique phospholipid profile of *C. benedicta*, which could indicate the specific adaptive or functional roles of these phospholipids in this species.

The low amount of lysophosphatidylcholine (<3.0%) observed in *C. benedicta* is consistent with the findings for *C. thracica*, indicating that this phospholipid is present in minimal quantities across different *Centaurea* species. Lysophosphatidylcholine is known for its role in inflammation and signaling, and its low abundance might reflect a conserved biochemical trait within the genus [12].

## 3. Materials and Methods

### 3.1. Materials

Seed material from the plant species *C. benedicta* (Blessed Thistle), Asteraceae family, was investigated. The seeds were collected from a selected plant, a collective population grown at the Institute of Rose and Essential Oil Crops (IREMC). The plants were collected in 2021. Then, the seeds were separated and subjected immediately to analysis.

### 3.2. Chemical Composition

The lipids were isolated by a Soxhlet apparatus with hexane for 8 h, and after that, the solvent was evaporated by a rotary vacuum evaporator [24]. A stream of nitrogen gas was used to remove the residual hexane. The content of oil was calculated.

The protein content was determined using a Kjeldahl apparatus (Velp Scientifica Srl, Via Stazione, Italy) after mineralization of the sample for 35 min at 420 °C in the presence of H_2_SO_4_:H_2_O_2_ (2:1, *v*/*v*) and a catalyst. The solution was distilled in UDK 127 [25].

The carbohydrate content was calculated as follows [26]:

100 − (weight in grams [protein + lipids + water + ash] in 100 g).

The fiber, ash, and moisture contents were determined gravimetrically [25].

The energy value (EV) in kcal/100 g was calculated as follows:

EV = %proteins × 17(4.0) + % carbohydrates × 17 (4.0) + lipids × 38 (9.0).

### 3.3. Physicochemical Properties

The physicochemical characteristics of the glyceride oil (iodine value, peroxide value, refractive index) were analyzed following previously published procedures [27,28,29]. The oxidative stability was measured on Rancimat 679 equipment (Metrohm, Herisau, Switzerland) at 100 °C [30].

### 3.4. Lipid Composition

#### 3.4.1. Fatty Acid Composition

The fatty acid composition of the seed oil was determined by gas chromatography (GC) [31] after trans-esterification of the glyceride oil with sulfuric acid in methanol in order to obtain fatty acid methyl esters (FAMEs) [32]. The determination of FAMEs was performed on an Agilent 8860 (Santa Clara, CA, USA) equipped with a flame ionization detector and capillary column (DB-Fast FAME, Agilent) with the following characteristics: 30 m × 0.25 mm × 0.25 μm (film thickness). The work conditions were as follows: the column temperature was 70 °C (for 1 min), up to 250 °C at a rate of 5 °C/min (hold for 3 min); the temperature of the injector was 270 °C, and that of the detector was 300 °C. Identification was performed by comparison of the retention times with the retention time of a standard mixture of FAMEs (Supelco, USA 37 comp. FAME mix) that was subjected to GC analysis under identical conditions.

#### 3.4.2. Sterols

A part of the seed oil (3 g) was saponified with 2 N KOH, and the unsaponifiable matter was extracted with hexane [33]. The total sterol content was determined spectrophotometrically at 597 nm after isolation by TLC [34]. The sterol composition was determined on an HP 5890 gas chromatograph equipped with a DB 5 capillary column (25 m × 0.25 mm × 0.25 µm (film thickness)) and FID. The operating conditions were as follows: temperature gradient from 90 °C (3 min) up to 290 °C at a rate of change of 15 °C/min and then up to 310 °C at a rate of 4 °C/min (10 min); detector temperature, 320 °C; injector temperature, 300 °C; and carrier gas is hydrogen. The individual composition of identified sterols was determined by a comparison of the retention times with a standard mixture of sterols [35].

#### 3.4.3. Tocopherols

The total content of tocopherols and individual tocopherols were determined by high-performance liquid chromatography (HPLC) on a Merck-Hitachi (Burladingen, Germany) with florescence detector F-1050 (Merck-Hitachi, Burladingen, Germany) under the following conditions: fluorescent detection (295 nm excitement and 330 nm emission) and Nucleosil Si 50-5 column (250 mm × 4 mm). The mobile phase was hexane/dioxane, 96:4 (*v*/*v*), and the flow rate was 1 mL/min [36]. Every tocopherol was identified by comparing the retention times with those of the standards (reference individual pure tocopherols—α-, β-, γ-, and δ-tocopherols with purity ≥ 98% from Merck (Darmstadt, Germany)). The tocopherol content was calculated on the basis of the tocopherol peak areas in the sample vs. the tocopherol peak area of a standard tocopherol solution.

#### 3.4.4. Phospholipids

The individual classes of phospholipids were isolated by two-dimensional thin-layer chromatography [37]. The individual phospholipids were detected and identified by spraying with specific reagents according to Christie [38]: Dragendorff test (detection of choline-containing phospholipids); Ninhydrin spray (for phospholipids with free amino groups); and Shiff’s reagent (for inositol-containing phospholipids). Additional identification was performed by comparing the respective Rf values with those of authentic commercial standards subjected to Silica gel 60 G TLC under identical experimental conditions. The spots of the individual phospholipids were scraped and mineralized with a mixture of perchloric and sulfuric acid, 1:1 (*v*/*v*). After that, the individual phospholipid contents were determined spectrophotometrically at 700 nm [39].

### 3.5. Statistics

All measurements were performed in triplicate (n = 3), and the results are presented as the mean value ± standard deviation (SD). IBM SPSS Statistics 25 was used for calculating the standard deviation, and One-Way ANOVA followed by a post hoc test (Duncan test) was performed for establishing the differences in the individual sterols.

## 4. Conclusions

The aim of this research was to determine whether the native Bulgarian species of *C. benedicta* from the Asteraceae family could be a potential source of nutrients and bioactive components. The seeds of *C. benedicta* were found to be rich in carbohydrates, especially dietary fibers, and had a relatively high protein content. Although the amount of glyceride oil was low, it was rich in oleic acid and essential linoleic acid. Additionally, it contained various lipid-soluble bioactive components, such as tocopherols (especially α-tocopherol), β-sitosterol, stigmasterol, phosphatidylethanolamine, and phosphatidylinositol. This composition suggests that *C. benedicta* seeds could be a valuable source of phytonutrients with applications in the human diet and various industries. However, to confirm this, a series of different tests must be performed: phytochemical analysis to establish the presence of bioactive compounds such as alkaloids, flavonoids, terpenoids, phenolic compounds, glycosides, and essential oils; toxicological analysis testing for harmful substances such as heavy metals, pesticides, and other contaminants to ensure safety for human consumption; microbiological analysis assessing the microbial load, including bacteria, fungi, and other pathogens, to ensure the plant or seed is safe for consumption; pharmacognostic analysis including morphological and anatomical studies to verify the identity of the plant species and examine its structural characteristics; and microscopic analysis to identify specific anatomical features that may contribute to its medicinal properties.

## Figures and Tables

**Figure 1 plants-13-03579-f001:**
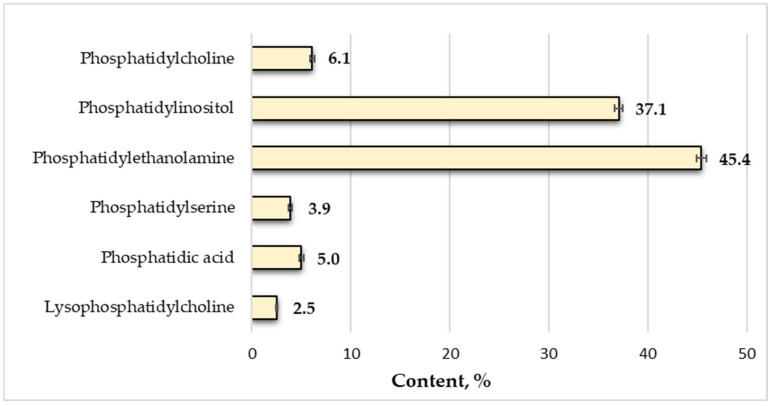
Individual phospholipid composition of *C. benedicta* seed oil.

**Table 1 plants-13-03579-t001:** Chemical composition of *C. benedicta* seeds.

Compound	Content in 100 g of Seeds	Content in 1000 Seeds
Oil content, % (d.w.)	11.0 ± 0.2	3.8 ± 0.1
Total proteins, % (d.w.)	16.4 ± 0.2	5.7 ± 0.1
Total carbohydrates, % (d.w.)	68.5 ± 0.5	23.8 ± 0.2
Fibers, % (d.w.)	32.2 ± 0.3	11.2 ± 0.1
Ash, % (d.w.)	4.1 ± 0.1	1.4 ± 0.0
Energy value, kcal/100 g (d.w.)	439	152

**Table 2 plants-13-03579-t002:** Physicochemical characteristics of *C. benedicta* seed oil.

Physicochemical Characteristic	Content
Peroxide value, meqO_2_/kg	2.8 ± 0.2
Iodine value, g I_2_/100 g	147.3 ± 0.2
Refractive index	1.4748 ± 0.0004

The samples were analyzed in triplicate (n = 3), and the results are expressed as mean ± standard deviation.

**Table 3 plants-13-03579-t003:** Content of biologically active components of *C. benedicta* seed oil.

Biologically Active Component	Content
Unsaponifiable matter, %	14.5 ± 0.4
Sterols, %	0.9 ± 0.1
Phospholipids, %	1.9 ± 0.1
Tocopherols, mg/kg	492 ± 82
α-Tocopherol, mg/kg	472 ± 78
β-Tocopherol, mg/kg	20 ± 4

The samples were analyzed in triplicate (n = 3), and the results are expressed as mean ± standard deviation.

**Table 4 plants-13-03579-t004:** Fatty acid composition of the total lipids from *C. benedicta* seeds.

Fatty Acid, %	Content, %
C _6:0_—Caproic acid	0.1 ± 0.0
C _8:0_—Caprylic acid	0.1 ± 0.0
C _13:0_—Tridecanoic acid	0.1 ± 0.0
C _14:0_—Myristic acid	0.1 ± 0.0
C _15:1_—Pentadecenoic acid	0.1 ± 0.0
C _16:0_—Palmitic acid	5.9 ± 0.1
C _16:1_—Palmitoleic acid C _17:0_—Margaric acid	0.1 ± 0.0 0.1 ± 0.0
C _17:1_—Heptadecenoic acid C _18:0_—Stearic acid	0.1 ± 0.0 2.6 ± 0.1
C _18:1_—Oleic acid	18.1 ± 0.2
C _18:2_ (n-6)—Linoleic acid	72.1 ± 0.4
C _18:3_ (n-3)—α-Linolenic acid	0.2 ± 0.0
C _20:1_—Eicosenoic acid	0.2 ± 0.0
C _22:0_—Behenic acid	0.1 ± 0.0
C _22:1_—Erucic acid	0.1 ± 0.0
C _22:6_ (n-3)—Docosahexaenoic acid	0.1 ± 0.0
Saturated fatty acids	9.0
Monounsaturated fatty acids	18.0
Polyunsaturated fatty acids	73.0

The samples were analyzed in triplicate (n = 3), and the results are expressed as mean ± standard deviation.

**Table 5 plants-13-03579-t005:** Individual sterol composition of *C. benedicta* seed oil.

Sterol	Content, %
Campesterol	7.5 ± 0.3 ^d^
Stigmasterol	19.4 ± 0.2 ^b^
β—Sitosterol	59.5 ± 0.5 ^a^
Δ^5^—Avenasterol	13.5 ± 0.3 ^c^
Δ^7^—Stigmasterol	0.1 ± 0.0 ^e^

The samples were analyzed in triplicate (n = 3), and the results are expressed as mean ± standard deviation. Different small letters in a column represent significant differences at *p* < 0.05 (Duncan test)

## Data Availability

The original contributions presented in this study are included in the article. Further inquiries can be directed to the corresponding author.

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
