# Peer review of "Centaurea benedicta*—A Potential Source of Nutrients and Bioactive Components"

_plants, 2024, doi:10.3390/plants13243579_

Round 1
Reviewer 1 Report
Comments and Suggestions for Authors
In her article "Centaurea benedicta - a potential source of nutrients and bioactive components" Olga Teneva and co-authors provide a comprehensive phytochemical analysis of the composition of Centaurea benedicta seeds in connection with their nutraceutical value. The authors provide a detailed composition of the key nutrients that make up the seeds of this plant. The work is done at a very good technical and methodological level and can even be considered exemplary in such studies, so I would like to recommend it for publication. At the same time, there are a number of points that should be clarified and specified, which, in my opinion, will improve the quality of the data presented.
First of all, I would like to receive detailed data on the absolute content of basic nutrients per gram of absolute dry weight, % water content in air-dried seeds and recalculation per 1 seed or 1000 seeds, which is important for agricultural technology. It seems somewhat strange that the authors provide some compounds in absolute values, and others in relative ones. Everything should be brought to a single format. What is the fractional composition of the extracted fatty oil? What does it contain besides triacylglycerols? What is the proportion of TAG, phospholipids, etc. in the oil?
There are also several questions about Table 4. First of all, a terminological correction - if the oil was not fractionated into separate lipid classes, then it is more correct to talk about the fatty acid composition of total lipids, and not call the fraction glyceride oil.
The second point, the table contains 6:0, 8:0 and then 14:0 - why are there no intermediate metabolites - acids formed as a result of C2 elongation of 8:0 acid? Most likely, the detection of C6-C8 acids is an artifact, they are not characteristic of lipids of the seeds of higher plants. As a rule, in addition to oleic acid, all higher plants contain cis-vaccenic acid in the composition of fatty oils. Was it detected in the samples? The detection of docosahexaenoic acid is an artifact — its biosynthesis in higher plants is impossible! An additional argument in favor of the artifact is that the sample does not contain intermediate metabolites of its biosynthesis — 18:4-20:5 acids. Figure 1 looks uninformative and unnecessary. It would be more logical to supplement Table 4 with the corresponding lines.
The last question concerns Figure 2 — the phospholipid classes are not labeled on it! There is also a question about the data. The authors write that they used two-dimensional TLC, it is known that this method is quite difficult to obtain good separation of phosphatidylinosine, phosphatidylcholine and phosphatidylserine — these classes either coelute or give a group of very close overlapping spots on TLC. As a rule, phosphatidylinosine is not the main phospholipid of seeds, it is minor. Most likely, the authors mistook phosphatidylcholine for it. To clarify the issue, it is strongly recommended to provide the results of two-dimensional TLC chromatography, developing one plate with ninhydrin, and staining the other one for choline using the Dragendorff method. The corresponding instructions can be found, for example, here:
https://cyberlipid.gerli.com/techniques-of-analysis/analysis-of-complex-lipids/phospholipid-analysis/tlc-identification-of-pl/
Author Response
Dear Editors and Reviewers,
We would like to extend our sincere gratitude for the thorough and insightful review of our manuscript titled "Centaurea benedicta – a potential source of nutrients and bioactive components" (Manuscript ID: plants-3389337) submitted to Plants. We appreciate the valuable comments and suggestions provided by the reviewers, which have significantly contributed to the improvement of our work.
Reviewer 1
In her article "Centaurea benedicta - a potential source of nutrients and bioactive components" Olga Teneva and co-authors provide a comprehensive phytochemical analysis of the composition of Centaurea benedicta seeds in connection with their nutraceutical value. The authors provide a detailed composition of the key nutrients that make up the seeds of this plant. The work is done at a very good technical and methodological level and can even be considered exemplary in such studies, so I would like to recommend it for publication. At the same time, there are a number of points that should be clarified and specified, which, in my opinion, will improve the quality of the data presented.
R1: First of all, I would like to receive detailed data on the absolute content of basic nutrients per gram of absolute dry weight, % water content in air-dried seeds and recalculation per 1 seed or 1000 seeds, which is important for agricultural technology.
A: We agree with the reviewer and corrected the text according to the suggestions. The results in Table 1 were precalculated and given as dry weight. The moisture content of the air-dried seeds was 5.4%. We also added a column where are given the values of the basic nutrient per 1000 seeds.
R1: It seems somewhat strange that the authors provide some compounds in absolute values, and others in relative ones. Everything should be brought to a single format.
A: Thank you for your comment. Most of the results were provided as percentages (%). However, the tocopherol results were given in mg/kg. We performed the analysis according to ISO 9936:2016 - Animal and Vegetable Fats and Oils. Determination of Tocopherol and Tocotrienol Contents by High-Performance Liquid Chromatography (ISO: Geneva, Switzerland, 2016), which requires the results to be presented in mg/kg.
R1: What is the fractional composition of the extracted fatty oil? What does it contain besides triacylglycerols? What is the proportion of TAG, phospholipids, etc. in the oil?
A: The content of the biologically active components in the glyceride oil is provided in Table 3, showing the percentages of unsaponifiable matter, sterols, and phospholipids. Unfortunately, we did not perform the fractional composition analysis of the lipids—mono-, di-, and triacylglycerols—so we are unable to present those results.
R1: There are also several questions about Table 4. First of all, a terminological correction - if the oil was not fractionated into separate lipid classes, then it is more correct to talk about the fatty acid composition of total lipids, and not call the fraction glyceride oil.
A: We agree with the reviewer and corrected the term into ‘total lipids.’
R1: The second point, the table contains 6:0, 8:0 and then 14:0 - why are there no intermediate metabolites - acids formed as a result of C2 elongation of 8:0 acid? Most likely, the detection of C6-C8 acids is an artifact, they are not characteristic of lipids of the seeds of higher plants.
A: These shorter-chain fatty acids are generally not characteristic of the lipids found in the seeds of higher plants. Higher plants primarily synthesize long-chain fatty acids through the process of fatty acid elongation, which occurs in the plastids. This involves a series of enzyme-catalyzed reactions that add two carbon units at a time to a growing fatty acid chain. Short-chain fatty acids are not typically produced or accumulated because the biosynthesis pathways favor the production of longer chains.
R1: As a rule, in addition to oleic acid, all higher plants contain cis-vaccenic acid in the composition of fatty oils. Was it detected in the samples?
A: Cis-vaccenic acid was not detected in the sample. Cis-vaccenic acid is not typical for the plants and is present in higher amounts in Sea buckthorns.
R1: The detection of docosahexaenoic acid is an artifact — its biosynthesis in higher plants is impossible! An additional argument in favor of the artifact is that the sample does not contain intermediate metabolites of its biosynthesis — 18:4-20:5 acids.
A: The detected content of docosahexaenoic acid (DHA) was 0.1%. The chromatogram revealed a peak at the retention time corresponding to the DHA standard.
R1: Figure 1 looks uninformative and unnecessary. It would be more logical to supplement Table 4 with the corresponding lines.
A: Corrected as suggested. We removed the Figure 1 and added the results in Table 4.
R1: The last question concerns Figure 2 — the phospholipid classes are not labeled on it!
A: Corrected as suggested. We added the labels of the phospholipids.
R1: There is also a question about the data. The authors write that they used two-dimensional TLC, it is known that this method is quite difficult to obtain good separation of phosphatidylinosine, phosphatidylcholine and phosphatidylserine — these classes either coelute or give a group of very close overlapping spots on TLC. As a rule, phosphatidylinosine is not the main phospholipid of seeds, it is minor. Most likely, the authors mistook phosphatidylcholine for it. To clarify the issue, it is strongly recommended to provide the results of two-dimensional TLC chromatography, developing one plate with ninhydrin, and staining the other one for choline using the Dragendorff method. The corresponding instructions can be found, for example, here:
https://cyberlipid.gerli.com/techniques-of-analysis/analysis-of-complex-lipids/phospholipid-analysis/tlc-identification-of-pl/
A: We appreciate the suggestion of the reviewer. Therefore, we added the full description for determination of the phospholipids:
‘The individual classes of phospholipids were isolated by two-dimensional thin-layer chromatography [37]. The individual phospholipids were detected and identified by spraying with specific reagents according to Christie [38]: Dragendorff test (detection of choline-containing phospholipids); Ninhydrin spray (for phospholipids with free amino groups), and Shiff’s reagent (for inositol containing phospholipids). Additional identifica-tion was performed by comparing the respective Rf values with those of authentic com-mercial standards subjected to Silica gel 60 G TLC under identical experimental condi-tions. The spots of the individual phospholipids were scrapped and mineralized with a mixture of perchloric and sulfuric acid, 1:1 (v/v). After that, the individual phospholipid content was determined spectrophotometrically at 700 nm [39].’
This was added in ‘3.4.4. Phospholipids’
Reviewer 2 Report
Comments and Suggestions for Authors
In this manuscript, the authors reported their findings on the phytochemical compositions of the seed of Centaurea benedicta. The results presented include nutrient contents, physicochemical properties of seed oil, as well as biologically active constituents encompassing fatty acids, sterols, tocopherols and phospholipids. While some interesting findings were presented, some parts of the manuscript could be improved further as described below. Discussion could also be made more significant/in-depth as suggested below. In the manuscript, “2. Results and discussion” is followed by “4. Materials and Methods”. “Materials and Methods” should be section 3 instead. Importantly, Figure 2 seems incomplete and should be rechecked.
1. ABSTRACT – This part can be improved by indicating at least briefly the knowledge gap being addressed by this study. Furthermore, for lines 21-26, it can be made more informative by indicating key findings, instead of just describing what were done.
2. Some statements are not supported by cited references as required. Please see the following and provide relevant references to support the information written:
· Lines 38-42: “It is native to… up to 1000 meters”.
· Lines 61-64: “C. benedicta is known for… and enhance diuresis”.
· Lines 109-110: “The lower number … of the oil”.
· Lines 143-145: “The results are in agreement with … the major component”.
3. All scientific names should be shown in italics. Please correct the ones in lines 60 (“Centaurea solstitalis”) and 80 (“Centaurea benedicta”).
4. By convention, after the first mention (“Centaurea benedicta”), subsequent mentions may take the form “C. benedicta”, where the genus is abbreviated. Please check lines 78, 80, and 83.
5. Please insert a space between “C.” and “benedicta”. Please check the ones to be corrected in lines 74, 148, and 156.
6. Please insert a space between “C.” and “kotschyi…”. Please check the ones to be corrected in lines 160 and 166.
7. For Table 1, it looks strange to have a hyphen before the word “Fibre”, please recheck whether it is a typo.
8. Line 111, please recheck whether “determinate” should be “determined” instead.
9. Line 155, please recheck whether “trade” should be “trait” instead.
10. Figure 1 seems redundant as the same information is repeated in the text in lines 178-183. If it must be included, I would recommend showing it in a table, or other form, where it is also possible to indicate the standard deviations of the data.
11. For the data in Table 5, it would be desirable to perform statistical tests, such as a one-way ANOVA followed by a post-hoc test, to rank the data so that the relative levels of the different constituents tested can be more objectively compared. This would also provide a more convincing interpretation of which component is the major one.
12. Importantly, please revise Figure 2. The figure is missing the information of the six individual phospholipid classes tested on the vertical axis. This is important. For Figure 2, it would be more informative to include standard deviation bars. Units for the data should be indicated too. It is also desirable to run statistical tests on the dataset as proposed above for Table 5. With that done, it is also possible to confirm whether the content of phosphatidylethanolamine was statistically significant (higher) compared to that of phosphatidylinositol.
13. Lines 242-225: Should “The plants...” be “The seeds ...” instead? Also, since the seeds were harvested three years ago in 2021, it would be good to at least briefly indicate how they were stored before they were used for analysis in this study.
14. For the “4.5 Statistics” – Was any software used? Anyway, if statistical tests are performed during revision as proposed above, please indicate the software used.
15. CONCLUSIONS
· Here the authors proposed that their results support the roles of the seeds and glyceride oil of C. benedicta in food and medicine. To support this proposal, the discussion (section 2) in the manuscript should be further strengthened to highlight any connections between the major constituents in the seed/oil of C. benedicta and any health/therapeutic benefits.
· In the current discussion, the authors have done a good job comparing the relative abundance of the phytoconstituents they tested with those reported in the literature. However, there is a lack of further explanation/interpretation on the importance of the differences found in those comparisons. The authors may consider elaborating those parts of the discussion, addressing how the findings can be linked to their stated aim of investigating whether C. benedicta seeds can serve “as a valuable and inexpensive source of phytonutrients” (line 76).
Author Response
Dear Editors and Reviewers,
We would like to extend our sincere gratitude for the thorough and insightful review of our manuscript titled "Centaurea benedicta – a potential source of nutrients and bioactive components" (Manuscript ID: plants-3389337) submitted to Plants. We appreciate the valuable comments and suggestions provided by the reviewers, which have significantly contributed to the improvement of our work.
Reviewer 2
R2: In this manuscript, the authors reported their findings on the phytochemical compositions of the seed of Centaurea benedicta. The results presented include nutrient contents, physicochemical properties of seed oil, as well as biologically active constituents encompassing fatty acids, sterols, tocopherols and phospholipids. While some interesting findings were presented, some parts of the manuscript could be improved further as described below. Discussion could also be made more significant/in-depth as suggested below.
A: We agree with the reviewer and modified the Discussion according to their suggestions.
In the manuscript, “2. Results and discussion” is followed by “4. Materials and Methods”. “Materials and Methods” should be section 3 instead. Importantly, Figure 2 seems incomplete and should be rechecked.
A: We appreciate the remarks of the reviewer and corrected the number of the sections according to the suggestion.
R2: ABSTRACT – This part can be improved by indicating at least briefly the knowledge gap being addressed by this study. Furthermore, for lines 21-26, it can be made more informative by indicating key findings, instead of just describing what were done.
A: We corrected the Abstract according to the suggestion.
“Abstract: The Asteraceae family is a large plant family, with over 1600 genera and 25 000 species, most of which are generally herbaceous plants. This family's members are widely used for the human diet and medicine. One of the most popular representatives is Centaurea benedicta L., known as ‘Blessed Thistle’. It is a famous plant in the herbal world with some medical benefits – strong antioxidant and antidepressant, with antibacterial and antiseptic properties, stimulant of the appetite, with a good effect on the liver and the secretion of bile juices etc. Therefore, this work aimed to fully characterize the chemical composition of the seeds of C. benedicta introduced in Bulgaria, some physicochemical characteristics as well as the biologically active compounds. The main nutrient in the chemical composition was carbohydrates (68.5%), as the half of their quantity was occupied by the fibers (32.2%). Total proteins were 16.4% and the glyceride oil content was rather low – about 11.0%. The main fatty acids identified in the seed oil was linoleic (72.1%) and oleic acids (18.1%) and the amount of the polyunsaturated ones predominated (73.0%). The main lipid-soluble bioactive components were sterols (0.9%), phospholipids (1.9%) and tocopherols (492 mg/kg). β-Sitosterol (59.5%) and stigmasterol (19.4%) were the main sterols and α-tocopherol (472 mg/kg) predominated in the tocopherol fraction. The major phospholipids were phosphatidylethanolamine (45.4%), followed by phosphatidylinositol (37.1%) and phosphatidylcholine (6.1%).”
R2: 2. Some statements are not supported by cited references as required. Please see the following and provide relevant references to support the information written:
- Lines 38-42: “It is native to… up to 1000 meters”.
- Lines 61-64: “C. benedicta is known for… and enhance diuresis”.
- Lines 109-110: “The lower number … of the oil”.
- Lines 143-145: “The results are in agreement with … the major component”.
A: We added the proper citation after all of the above-mentioned lines.
- Lines 38-42: “It is native to… up to 1000 meters”. – [3]
- Lines 61-64: “C. benedicta is known for… and enhance diuresis”. – [6, 7]
- Lines 109-110: “The lower number … of the oil”. – [15]
- Lines 143-145: “The results are in agreement with … the major component”. – [18]
R2: 3. All scientific names should be shown in italics. Please correct the ones in lines 60 (“Centaurea solstitalis”) and 80 (“Centaurea benedicta”).
A: Corrected as suggested.
R2: 4. By convention, after the first mention (“Centaurea benedicta”), subsequent mentions may take the form “C. benedicta”, where the genus is abbreviated. Please check lines 78, 80, and 83.
A: Corrected as suggested.
R2: 5. Please insert a space between “C.” and “benedicta”. Please check the ones to be corrected in lines 74, 148, and 156.
A: Corrected as suggested.
R2: 6. Please insert a space between “C.” and “kotschyi…”. Please check the ones to be corrected in lines 160 and 166.
A: Corrected as suggested.
R2: 7. For Table 1, it looks strange to have a hyphen before the word “Fibre”, please recheck whether it is a typo.
A: Corrected as suggested.
R2: 8. Line 111, please recheck whether “determinate” should be “determined” instead.
A: Corrected as suggested.
R2: 9. Line 155, please recheck whether “trade” should be “trait” instead.
A: We corrected it as follow: “This pattern was also observed in the fatty acid composition of our sample.”
R2: 10. Figure 1 seems redundant as the same information is repeated in the text in lines 178-183. If it must be included, I would recommend showing it in a table, or other form, where it is also possible to indicate the standard deviations of the data.
A: Corrected as suggested. We removed the Figure 1 and added the results in Table 4.
R2: 11. For the data in Table 5, it would be desirable to perform statistical tests, such as a one-way ANOVA followed by a post-hoc test, to rank the data so that the relative levels of the different constituents tested can be more objectively compared. This would also provide a more convincing interpretation of which component is the major one.
A: One-Way ANOVA followed by a post-hoc test (Duncan test) was performed for establishing the differences in the individual sterols (IBM SPSS Statistics 25).
R2: 12. Importantly, please revise Figure 2. The figure is missing the information of the six individual phospholipid classes tested on the vertical axis. This is important. For Figure 2, it would be more informative to include standard deviation bars. Units for the data should be indicated too. It is also desirable to run statistical tests on the dataset as proposed above for Table 5. With that done, it is also possible to confirm whether the content of phosphatidylethanolamine was statistically significant (higher) compared to that of phosphatidylinositol.
A: The figure was corrected as the reviewer’s suggestions.
R2:13. Lines 242-225: Should “The plants...” be “The seeds ...” instead? Also, since the seeds were harvested three years ago in 2021, it would be good to at least briefly indicate how they were stored before they were used for analysis in this study.
A: We corrected the text as follows: “The plants were collected in 2021. Then, the seeds were separated and subjected immediately to analysis.”
R2: 14. For the “4.5 Statistics” – Was any software used? Anyway, if statistical tests are performed during revision as proposed above, please indicate the software used.
A: The following text was added: “IBM SPSS Statistics 25 was used for calculating the standard deviation and One-Way ANOVA followed by a post-hoc test (Duncan test) was performed for establishing the differences in the individual sterols.”
- CONCLUSIONS
R2: Here the authors proposed that their results support the roles of the seeds and glyceride oil of C. benedicta in food and medicine. To support this proposal, the discussion (section 2) in the manuscript should be further strengthened to highlight any connections between the major constituents in the seed/oil of C. benedicta and any health/therapeutic benefits.
- In the current discussion, the authors have done a good job comparing the relative abundance of the phytoconstituents they tested with those reported in the literature. However, there is a lack of further explanation/interpretation on the importance of the differences found in those comparisons. The authors may consider elaborating those parts of the discussion, addressing how the findings can be linked to their stated aim of investigating whether C. benedictaseeds can serve “as a valuable and inexpensive source of phytonutrients” (line 76).
A: We agree with the reviewer’s remark that based only on the examined chemical and lipid composition we are not able to draw conclusion on the health-benefits on the plant seeds. For that reason, we changed the Conclusion to:
“The aim of this research was to determine whether the native Bulgarian species of C. benedicta from the Asteraceae family could be a potential source of nutrients and bioactive components. The seeds of C. benedicta were found to be rich in carbohydrates, especially dietary fibers, and had a relatively high protein content. Although the amount of glyceride oil was low, it was rich in oleic acid and essential linoleic acid. Additionally, it contained various lipid-soluble bioactive components, such as tocopherols (especially α-tocopherol), β-sitosterol, stigmasterol, phosphatidylethanolamine, and phosphatidylinositol. This composition suggests that C. benedicta seeds could be a valuable source of phytonutrients with applications in the human diet and various industries. However, to confirm this, a series of different tests must be performed: phytochemical analysis to establish the presence of bioactive compounds such as alkaloids, flavonoids, terpenoids, phenolic compounds, glycosides, and essential oils; toxicological analysis testing for harmful substances such as heavy metals, pesticides, and other contaminants is crucial to ensure safety for human consumption; microbiological analysis assessing the microbial load, including bacteria, fungi, and other pathogens, to ensure the plant or seed is safe for consumption; pharmacognostic analysis including morphological and anatomical studies to verify the identity of the plant species and examine its structural characteristics; microscopic analysis to identify specific anatomical features that may contribute to its medicinal properties.”
Reviewer 3 Report
Comments and Suggestions for Authors
Centaurea benedicta is a well-known medicinal herb with key pharmacological properties including antimicrobial, antioxidant, antidepressants, among others.
In the present era of drug discovery, bio-based resources are extensively explored and optimized for their therapeutic functions. Medicinal plants have been widely documented for their potent efficacies and pharmacological functions.
I have a few major and minor suggestions for the improvement of the manuscript.
The manuscript is of research significance since very few studies are performed on the plant- please refer to Sousa et al. 2021, Cnicin for skin inflammation; Kahraman et al, 2025, antihyperglycemic potential of seven species, etc. The complete characterization of seed chemical composition in the plant would be of value for alternative medicines and food and also for researchers working on plant bioactive compounds.
Among different plant parts, why plant seeds were chosen for the study? Considering the plant remains less studied. Was there any reason for selecting seeds? Discuss.
Line 84: The oil content of the plant species……was determined….is it the entire plant or only the seed oil composition?
Line 84-85, it is mentioned that the oil content was found to be almost half for the same plant reported in other studies? What conclusion can be drawn?
The fatty acid composition of C. benedicta seed oil showed a higher content of Linoleic acid, Oleic acid partially Palmitic acid, others are less present.
The details of the instruments (GC and HPLC) should be included in the material and method section (name, manufacturer etc.)
How can the present study contribute to our understanding and what are the future directions? While characterization of seed composition is not enough for its use in food or medicine. Some components may demonstrate adverse impact on human body on consumption? What is author’s point of view?
A brief discussion on the limitations and future optimization of the study is necessary.
Minor comments
Line 80: Centaurea benedicta should be in italics and likewise in the manuscript.
Extensive English revision is required, e.g. line 84-85, The amount was almost half of the reported………the sentence needs to be revised for clarity.
Comments on the Quality of English Language
The paper should be extensively revised for English language.
Author Response
Dear Editors and Reviewers,
We would like to extend our sincere gratitude for the thorough and insightful review of our manuscript titled "Centaurea benedicta – a potential source of nutrients and bioactive components" (Manuscript ID: plants-3389337) submitted to Plants. We appreciate the valuable comments and suggestions provided by the reviewers, which have significantly contributed to the improvement of our work.
Reviewer 3
Centaurea benedicta is a well-known medicinal herb with key pharmacological properties including antimicrobial, antioxidant, antidepressants, among others.
In the present era of drug discovery, bio-based resources are extensively explored and optimized for their therapeutic functions. Medicinal plants have been widely documented for their potent efficacies and pharmacological functions.
I have a few major and minor suggestions for the improvement of the manuscript.
The manuscript is of research significance since very few studies are performed on the plant- please refer to Sousa et al. 2021, Cnicin for skin inflammation; Kahraman et al, 2025, antihyperglycemic potential of seven species, etc. The complete characterization of seed chemical composition in the plant would be of value for alternative medicines and food and also for researchers working on plant bioactive compounds.
R3: Among different plant parts, why plant seeds were chosen for the study? Considering the plant remains less studied. Was there any reason for selecting seeds? Discuss.
A: The primary reason for focusing on the seeds in this study was to examine the main nutrients, particularly the lipid-soluble bioactive components. Since glyceride oil is concentrated in the seeds, we chose to work with them.
R3: Line 84: The oil content of the plant species……was determined….is it the entire plant or only the seed oil composition?
A: We change it to: “The oil content of the seeds from C. benedicta was determined to be 11.0%.”
R3: Line 84-85, it is mentioned that the oil content was found to be almost half for the same plant reported in other studies? What conclusion can be drawn?
A: These variations could be attributed to the different climatic conditions under which the plants were grown. Additionally, the type of soil and the annual precipitation rate can also significantly influence the biosynthesis of glyceride oils in plant cells.
R3: The fatty acid composition of C. benedicta seed oil showed a higher content of Linoleic acid, Oleic acid partially Palmitic acid, others are less present.
A: The obtained results for the fatty acid composition of the C. benedicta seed oil was close to the findings of Ghiasy-Oskoee and Agha Alikhani, 2023.
R3: The details of the instruments (GC and HPLC) should be included in the material and method section (name, manufacturer etc.)
A: The details of the GC instrument were given in Section 3.4.1. Fatty acid composition: Agilent 8860 (Santa Clara, CA, USA) equipped with a flame ionization detector and capillary column (DB-Fast FAME, Agilent) with the following characteristics: 30 m × 0.25 mm × 0.25 μm (film thickness).
The details of the HPLC were added in Section 3.4.3. Tocopherols: Total content of tocopherols and individual tocopherols were determined by high-performance liquid chromatography (HPLC) on Merck-Hitachi (Burladingen, Germany) with florescence detector F-1050 (Merck-Hitachi, Burladingen, Germany) under the following conditions: fluorescent detection (295 nm of excitement and 330 nm of emis-sion) and Nucleosil Si 50-5 column (250 mm × 4 mm).
R3: How can the present study contribute to our understanding and what are the future directions? While characterization of seed composition is not enough for its use in food or medicine. Some components may demonstrate adverse impact on human body on consumption? What is author’s point of view?
A: We agree with the reviewer’s remark that based only on the examined chemical and lipid composition we are not able to draw conclusion on the health-benefits on the plant seeds. For that reason, we changed the Conclusion to:
“The aim of this research was to determine whether the native Bulgarian species of C. benedicta from the Asteraceae family could be a potential source of nutrients and bioactive components. The seeds of C. benedicta were found to be rich in carbohydrates, especially dietary fibers, and had a relatively high protein content. Although the amount of glyceride oil was low, it was rich in oleic acid and essential linoleic acid. Additionally, it contained various lipid-soluble bioactive components, such as tocopherols (especially α-tocopherol), β-sitosterol, stigmasterol, phosphatidylethanolamine, and phosphatidylinositol. This composition suggests that C. benedicta seeds could be a valuable source of phytonutrients with applications in the human diet and various industries. However, to confirm this, a series of different tests must be performed: phytochemical analysis to establish the presence of bioactive compounds such as alkaloids, flavonoids, terpenoids, phenolic compounds, glycosides, and essential oils; toxicological analysis testing for harmful substances such as heavy metals, pesticides, and other contaminants is crucial to ensure safety for human consumption; microbiological analysis assessing the microbial load, including bacteria, fungi, and other pathogens, to ensure the plant or seed is safe for consumption; pharmacognostic analysis including morphological and anatomical studies to verify the identity of the plant species and examine its structural characteristics; microscopic analysis to identify specific anatomical features that may contribute to its medicinal properties.”
R3: A brief discussion on the limitations and future optimization of the study is necessary.
A: The limitations and future optimization of the study were added in the Conclusion.
Minor comments
R3: Line 80: Centaurea benedicta should be in italics and likewise in the manuscript.
A: Corrected as suggested.
R3: Extensive English revision is required, e.g. line 84-85, The amount was almost half of the reported………the sentence needs to be revised for clarity.
A: The manuscript underwent extensive English revision to enhance its quality.
Round 2
Reviewer 3 Report
Comments and Suggestions for Authors
The manuscript has been improved as per the suggestions and can be considered for publication.